# Distinctive Properties of Endothelial Cells from Tumor and Normal Tissue in Human Breast Cancer

**DOI:** 10.3390/ijms22168862

**Published:** 2021-08-17

**Authors:** Kinga Wilkus, Klaudia Brodaczewska, Arkadiusz Kajdasz, Claudine Kieda

**Affiliations:** 1Laboratory of Molecular Oncology and Innovative Therapies, Military Institute of Medicine, PL-04-141 Warsaw, Poland; kbrodaczewska@wim.mil.pl (K.B.); ckieda@wim.mil.pl (C.K.); 2Postgraduate School of Molecular Medicine, Medical University of Warsaw, PL-02-091 Warsaw, Poland; 3Laboratory of Human Molecular Genetics, Faculty of Biology, Institute of Molecular Biology and Biotechnology, Adam Mickiewicz University Poznan, 61-614 Poznan, Poland; arkadiusz.kajdasz@amu.edu.pl; 4Center for Molecular Biophysics UPR 4301 CNRS, 45071 Orleans, France

**Keywords:** microenvironment, angiogenesis, endothelial cells, breast cancer, organospecificity, vascular dysfunction

## Abstract

Tumor microenvironments shape aggressiveness and are largely maintained by the conditions of angiogenesis formation. Thus, endothelial cells’ (ECs) biological reactions are crucial to understand and control the design of efficient therapies. In this work, we used models of ECs to represent a breast cancer tumor site as well as the same, healthy tissue. Cells characterization was performed at the transcriptome and protein expression levels, and the cells functional biological responses (angiogenesis and permeability) were assessed. We showed that the expression of proteins specific to ECs (ACE+, VWF+), their differentiation (CD31+, CD 133+, CD105+, CD34-), their adhesion properties (ICAM-1+, VCAM-1+, CD62-L+), and their barrier formation (ZO-1+) were all downregulated in tumor-derived ECs. NGS-based differential transcriptome analysis confirmed CD31-lowered expression and pointed to the increase of Ephrin-B2 and SNCAIP, indicative of dedifferentiation. Functional assays confirmed these differences; angiogenesis was impaired while permeability increased in tumor-derived ECs, as further validated by the distinctly enhanced VEGF production in response to hypoxia, reflecting the tumor conditions. This work showed that endothelial cells differed highly significantly, both phenotypically and functionally, in the tumor site as compared to the normal corresponding tissue, thus influencing the tumor microenvironment.

## 1. Introduction

As breast cancer is one of the major causes of death among the female population worldwide, numerous studies are carried on breast tumor angiogenesis [1]. In the great majority of them, human umbilical vein endothelial cells (HUVECs) were used as the universal model of endothelial cells (ECs). However, such a widespread use may lead to skewed observations due to the heterogeneity of donors and changes in phenotype caused by long-term culture [2]. Furthermore, they do not represent, by any means, the organs in which cancers originate. The actual, recognized organospecificity of endothelial cells has not been considered. Moreover, the pathologic conditions of the tumor microenvironment are neglected, what is another source of misinterpretation of results. Therefore, setting up a relevant experimental models for angiogenesis is crucial to better understanding the molecular mechanisms triggered during tumor growth and metastasis and to provide diagnostic, prognostic and treatment opportunities.

Endothelial cells form monolayer in all blood vessel walls and fulfill multiple functions, helping to maintain vascular homeostasis, regulate blood flow and blood clotting, control vessel wall permeability and regulate proper trafficking and recirculation of leukocytes [3,4]. Besides its physiological function, the endothelium plays an important role in processes that occur during the progression of diseases particularly those affecting the vascular system. Angiogenesis is a fundamental mechanism in cancer development. In the hypoxic microenvironments of tumors, endothelial cells perform angiogenesis, leading to the abnormal structure and function of the blood vessels, which further maintains hypoxia instead of compensating for it [5]. The biological differences between tumor endothelial cells (TECs) and normal endothelial cells (NECs) in tissues or organs add to the heterogeneity that must be considered for endothelial cell properties between organs and vessel classes in view of data interpretations [6].

Endothelial cells exhibit numerous characteristic markers that define the cell type. Endothelial cells present von Willebrand factor (VWF) [7] and platelet endothelial cell adhesion molecule 1 (PECAM-1, CD31) [8]. Moreover, ECs, on their surface, present endoglin (CD105), which regulates their proliferation [9]. Considering their differentiation steps, known endothelial progenitor cells (EPCs) are CD133+, CD34+, CD44+ and CD202b+. This set of markers is a combination that evolves according to maturation and specialization, leading to organospecificity. Considering the vessel type, lymphatic ECs are characterized by the presence of VEGFR3 [10], podoplanin and lymphatic vessel endothelial receptor 1 (LYVE-1). Venous ECs present Ephrin-B4 receptor tyrosine kinase, whereas arterial endothelial cells exhibit Ephrin-B2 [11]. After stimulation by tumor necrosis factor (TNF), bacterial lipopolysaccharide (LPS) and interleukin-1 (IL-1), ECs express cell adhesion molecules such as intercellular cell adhesion molecule 1 (ICAM-1, CD54) [12], vascular endothelial cell adhesion molecule 1 (VCAM-1, CD106) [13] and E-selectin (E-Sel, CD62-E) [14] on the cell surface. n their surfaces, ECs also express CD309 (VEGFR-2, KDR-1), which controls cell proliferation and migration and may modulate endothelial permeability [15,16]. Endothelial cells exhibit angiotensin converting enzyme activity (ACE; CD143), involved in the metabolism of angiotensin [17] and the inactivation of bradykinin. In functional assays, endothelial cells are able to form pseudo-vessels in vitro on Matrigel^TM^ as a mimicry of the extracellular matrix [18,19].

Herein, we showed that newly established cell lines, organo-specifically representing normal healthy breast tissue (HBH.MEC (healthy ECs)) and tumor site-derived endothelial cells (HBCa.MEC), though isolated from the same breast tumor patient, differed from one another. Our study presented new insights into the phenotype and established differences between healthy and pathological breast endothelia for valid mimicry of breast tumor conditions. Our data validated established endothelial cell lines that maintained their EC character during long-term culture as more relevant tools than those generally known and used [20,21].

## 2. Results

### 2.1. Tumor Microenvironment Influences EC Morphology, Proliferation and Expression of BCL6/p53

Both cell lines maintained their cobblestone morphology in monolayer cultures along numerous passages (Figure 1A). Healthy tissue-derived ECs showed a tendency to display a higher proliferation rate in standard culture conditions than breast tumor-derived ECs (Figure 1B,C). The level of BCL6 protein, a negative regulator of apoptosis, was lower in pathological ECs than in healthy ECs, whereas p53 protein levels displayed an opposite tendency (Figure 1D). 

### 2.2. HBH.MEC and HBCa.MEC Lines Display ECs Phenotype

Cells were stained for endothelial cell-specific markers (CD31, VWF, ACE, CD133, CD34, CD105) and related to pathological states of tissue (PDPN, AP, αSMA, EGFR). The unstained cells were used to test for autofluorescence and unconjugated primary antibodies (VWF, ACE). Controls were set using secondary labeled antibodies for nonspecific labeling levels. 

The mean fluorescence of cells for each marker is presented on graphs and representative histograms are shown (Figure 2). In general, healthy ECs and tumor-derived ECs were positive for all tested markers except for CD34 (VWF+, ACE+, CD31+, CD133+, CD105+), but pathological ECs were characterized by lower levels of expression (Figure 2). Moreover, both cell lines expressed the adhesion molecules ICAM-1, VCAM-1 and CD62L+. ICAM displayed a similar level of expression in both cell lines (Appendix A). In addition, both cell lines expressed PDPN, EGFR and AP with higher levels of expression of these markers in healthy ECs than in pathological ECs. PDPN expression, however, showed a certain degree of variability. The entire tumorous EC population showed higher levels of αSMA and a tendency to express EGFR more effectively than healthy ECs (Figure 2B). 

### 2.3. Hypoxia Induces the Production of VEGF-A by HBH.MEC and HBCa.MEC

To determine how the microenvironment influenced VEGF-A secretion by mature ECs, cells were cultured under normoxia or hypoxia for 48 h. Then, the level of VEGF-A was measured in medium. The production of VEGF-A was stimulated upon hypoxia in the case of both cell lines, with significantly higher rates and differential expressions in pathological ECs than in healthy ECs (Figure 3). Moreover, tumor-derived ECs secreted a lower levels of VEGF-A in normoxia (Figure 3).

### 2.4. Proangiogenic Response In Vitro Depends on the Pathological State of ECs

To observe the functional activity of the studied cells, tube formation assay was performed. Healthy and tumor-derived ECs were assessed for their angiogenic potential on Matrigel^TM^-coated plates. Both cell lines were able to perform pseudotube formation, as previously shown [22]. In the case of HBH.MEC, angiogenesis was achieved after 5 h, whereas HBCa.MEC created networks but did not achieve fully-formed tubes (Figure 4A). Healthy ECs presented a higher number of both nodes and junctions than tumor-derived ECs (Figure 4B). We also checked the levels of CD309 (FLK1), which induces angiogenesis as well as permeabilization of cell monolayers, in both cell lines (Appendix A). Although both cell lines were CD309+, the levels of vascular endothelial growth factor receptor 2 were higher in healthy ECs than in HBCa.MEC cultured in normoxia (Figure 4B).

### 2.5. The Transcriptome Analysis of Breast Tumor-Derived ECs Compared with Healthy ECs Revealed a Distinct Expression Profile Implicated in Vascular Development and Angiogenesis

Validation of the organo- and biological significance of the differences between normal and pathological ECs in the same organ was performed. Transcriptome comparison of healthy and tumor-derived ECs demonstrated significantly differentially expressed genes. as shown in the graph, the red and blue dots represent significantly up-regulated and down-regulated genes, respectively (Figure 5A). NGS data indicated for 350 up-regulated genes and 396 down-regulated genes. We chose 279 genes with logFC > or < ±1.5 and *q* value < 0.05. Among these, we found the ten with the most-changed mRNA levels between both cell lines (Figure 5B). Then, analysis was performed in Cytoscape (v.3.8.0) in order to assess the protein–protein interaction network of ECs’ differentially expressed molecules. The interaction network showed nodes representing PECAM-1 downregulation and Ephrin-B2 upregulation; these are EC markers and are significant for the differentiation state (Figure 5C). SNCAIP suggested a strong relation to tumoral plasticity. To assess the transcriptomic characteristics based on gene ontology, we selected the 279 most-expressed genes per cell type, based on UniProt analysis. We detected enrichment of multiple vasculature-related biological processes, as well as processes related to muscle differentiation. The 10 most activated processes per cell type are shown in Figure 5D.

## 3. Discussion

Due to the heterogeneity of ECs, it is extremely important to use ECs that are as similar as possible to in vivo conditions in order to better mimic the cancer microenvironment. We proposed new cell lines as a model for angiogenesis studies in breast cancer biology, as breast cancer is one of the common cancer types among women worldwide [1]. The proposed ECs are human, mature, organospecific, immortalized cell lines and were isolated from the same patient, which decreases variability between donors. In this paper, we focused on the characterization and comparative analysis of these cell lines in terms of cell growth, morphology, cellular markers, gene expression patterns and their functionality in vitro.

After immortalization, as patented, and many passages, the ECs presented the endothelial phenotype and clearly displayed biologically and phenotypically significant differences. Both cell lines were positive for ACE, the criterion that kept the previously-immortalized human endothelial cell lines [20]. Additionally, these ECs expressed another endothelial marker, CD105, similarly to those studied by Grange et al. [23]. Importantly, we found that both ECs lines presented main markers of differentiated endothelial cells: VWF and CD31 [24]. PECAM-1 was also identified in NGS as being downregulated in breast TECs vs. breast NECs, which underlines the fact that the tumor microenvironment alters the EC phenotype. The characteristic surface marker expression pattern was highly representative of endothelial cells. At the same time, the cell lines were negative for CD34 and positive for CD133, with the latter being a marker of not only endothelial progenitor cells, but also hematopoietic lineage, as shown by Ohga et al. [25]. The expression of CD133 was previously detected in high (but not low) metastatic tumor blood vessels [25]. Nevertheless, the present cells’ surface phenotype confirmed their endothelial origin, excluding a hematopoietic lineage evolution as well as their response to hypoxia. As Paprocka et al. observed, exposure to hypoxia induced VEGF-A secretion in both cell lines [26]. Although tumor ECs produced less VEGF than healthy EC at a basal level, the level of induction in response to hypoxia was much stronger. This reflected the pathologic angiogenic response and explained the ineffectiveness of the tumor vessels as well as their permeability and inability to alleviate hypoxia.

ECs derived from healthy breast tissue displayed higher proliferation rates- measured by metabolic activity- than tumor–derived ECs. They also expressed higher BCL6 protein levels. Additionally, p53 appeared to be less active in these cells, contrary to breast-tumor derived ECs. This could indicate that the tumor microenvironment, from which the ECs were isolated, had a prolonged influence on their growth and the expression of proapoptotic proteins. The cells’ origin also affected also their proangiogenic potential. Indeed, the obtained endothelial cell lines could form capillary-like structures in Matrigel^TM^ coated plates, which is a feature of mature ECs and not of undifferentiated breast tumor progenitor cells [23]. Both of our cell lines were able to form vessels in vitro, but the efficiency of this process depended on their origin. HBCa.MEC achieved a complete network in the studied timeframe, showing lower angiogenic potential than healthy ECs. Moreover, both cell lines were CD309+, with higher expressions of CD309 in healthy ECs. Newly formed networks, initiated by VEGF-A/VEGFR2, resulted in a higher rates of healthy EC proliferation, survival and new vessel formation than tumor-derived ECs [27]. This phenomenon was partly observed in present study, as the permeability assay indicated that tumor-derived EC monolayers leaked more than healthy EC monolayers. This distinct permeability may have occurred due to the lower ZO-1 protein level in pathologic cells. In our study, we detected a lower expression of alkaline phosphatase in cancer ECs. This protein is largely characteristic of blood–brain barrier derived ECs but has been shown to increase upon re-induction of barrier properties [28,29]. This may further suggest a defective angiogenesis regulation in the tumor endothelial cell line.

Our cell lines also displayed differences in leukocyte rolling-related protein levels such as ICAM-1, PECAM-1, VCAM-1, which were expressed on the cell surface as on the nonlymphoid tissue–derived microvascular ECs, HIMEC.1 and HSKMEC.1 [20,27]. CD54 was less expressed in breast tumor-derived ECs than in healthy ones, corroborating the phenotype of tumor ECs vs. the normal features of healthy ECs. VCAM-1 and ICAM-1, which mediate leukocyte—endothelial cell adhesion, were expressed in our model [30]. Furthermore, the presence of L-selectin—the receptor responsible for the initial steps of leukocyte rolling [31]—suggested that the obtained cell lines could be used as a model for endothelium—immune cell interaction [32].

Apart from EC-specific markers, we also evaluated proteins related to of the ECs’ origin and tumor-related proteins. Both of our cell lines were PDPN+, a controversial marker for both arterial and lymphatic ECs, as Furukoji et al. and Hatakeyama et al. showed [33,34]. The expression of EphrinB2 served as further evidence that both ECs originated from the arteries [11], as shown by NGS. Additionally, this gene has also been associated with poor prognosis in HER2-positive breast cancer [35] and has been up-regulated in breast TECs. Moreover, NGS analysis indicated upregulation of SNCAIP in tumor-derived ECs, which is almost exclusively expressed in triple-negative breast tumors (protein atlas), suggesting the tumor-related phenotype of the obtained cell lines. Another tumor-related marker that is up-regulated in pathological ECs vs. healthy ECs is ABCG2, a multidrug-resistance receptor (MDR) in breast cancer cells [36] that is also expressed by various ECs [37,38].

Moreover, pathologic ECs tend to present increased level of mesenchymal markers (α-SMA) than healthy ones. SMA is used to identify vascular smooth muscle cells and pericytes [39] but is also present on arterioles/venules, rather than capillaries [40]. However, some endothelial cells, especially in in vitro culture, have also been shown to express this marker [41]. Importantly, SMA was shown to increase in vessels upon inflammation and during fibrosis, which confirms the pathological features of cancer-derived ECs [42].

Global gene analysis identified several genes that showed differential expression in healthy and tumor-derived ECs. The 10 most changed expressed genes in cancer-derived EC were previously shown to play roles in cancer cells. Upregulated genes in pathological ECs vs. healthy ECs include ABCG2, also known as breast cancer resistance protein (BCRP) and lipid phosphate phosphatase-related protein type 5 (PLPR5), found in lung cancer and in breast tumors [36,43]. Melanoma-associated antigen 11 (MAGAB) is expressed in several types of tumors, such as melanoma, head and neck squamous cell carcinoma, lung carcinoma and breast carcinoma [44]. Overexpression of MORN4 is found in breast cancer tissue [45]. SNCAIP mutation is mostly found in cutaneous melanoma but also occurs in breast invasive ductal carcinoma [46]. Additionally, O51B4, which plays a role in olfactory receptors and in some cancers, was decreased in TECs [47]. The downregulated genes specific to cancer ECs were characterized before dropping in tumor cells. The genes with lower expression in HBCa.MEC are related to AJAP1(a tumor suppressor), HBG2 (down-regulated in ovarian cancer) [48] and CBPE(a modulator of actin filaments’ organization) [49]. Apparently, tumor-derived ECs display gene expression patterns characteristic of both endothelium and cancer cells.

## 4. Materials and Methods

### 4.1. Endothelial Cell Lines Culture

Endothelial cells were established according to the method previously described [20,21] (CNRS patent 99-16169). The samples were obtained from a female patient (INSERM UMR 1186, Integrative Tumor Immunology and Genetic Oncology, Gustave Roussy, EPHE, Villejuif, France), diagnosed with breast cancer (stage IIA: T2-N1-M0; HR-/HER2-). Resection specimens of primary tumor and healthy tissue, were received freshly after surgery, with informed written patient consent. All procedures were performed in accordance with generally accepted guidelines for the use of human material. The samples of healthy tissue and primary breast tumor were named HBH.MEC and HBCa.MEC, respectively. HBH.MEC and HBCa.MEC were seeded at density 5 × 10^4^ cells/10cm2. Both ECs lines were cultured on Primaria Tissue Culture Flask (Corning, NY, USA, #353808) in the presence of Opti-MEM I Reduced Serum Medium (Gibco, Paisley, UK #31985070) supplemented with 2% (*vol/vol*) fetal bovine serum (Gibco, Paisley, UK #A3840402). All cells were maintained at 37 °C and 5% CO_2_ in humidified atmosphere. Lastly, prior to experiments, cells were detached with enzyme cell detachment medium: Accutase supplied in Dulbecco’s PBS containing 0.5 mM EDTA and phenol red (Invitrogen, Carlsbad, CA, USA #E136579). The cell lines were mycoplasma-free (PromoKine, Hamburg, Germany #PK-CA91-1096).

### 4.2. Cell Viability Assay

HBH.MEC and HBCa.MEC, were seeded on 96-well plates, 1500 cells/well. Cells were cultured in 200 µL of normoxic medium for 48 h. Then, to assess cell viability, Alamar Blue Cell Viability Reagent (Invitrogen, Carlsbad, CA, USA #DAL1100) was added to the wells and incubated for 4 h at 37 °C. After this time, the absorbance of alamarBlue was read at 570 nm against the blank established by cell-free wells filled with medium.

### 4.3. Flow Cytometry Analysis

After the cells were collected and washed twice with PBS, they were incubated with the recommended dilution of antibodies. The samples were stored at 4 °C for 30 min and washed with PBS. Then, cells were analyzed by flow cytometry using CYTOFLEX software v.2.3.0.84 (Becton Dickinson, Franklin Lakes, NJ, USA). The lower threshold was used to exclude debris and live cells with gating (10,000 cells), according to forward scatter (FSC) × side scatter (SSC), followed by sections containing antibodies. The following antihuman antibodies were used: APC-conjugated EGFR Antibody (BioLegend, San Diego, CA, USA #352905), PerCP-conjugated Podoplanin Antibody (eBioscience, San Diego, CA, USA #46-9381-42), PE-Cy7-conjugated CD34 Antibody (BD Biosciences, Franklin Lakes, NJ, USA #560710), PE-conjugated CD133 (Miltenyi Biotec, Bergisch Gladbach, Germany#130-098-046), FITZ-conjugated CD105 (BioLegend, San Diego, CA, USA #323204), PE-conjugated Anti-CD309 Antibody (Beckman Coulter Life Sciences, Marseille Cedex, France #a64615), Alexa Fluor 700-conjugated BCL-6 Antibody (BD Biosciences, Franklin Lakes, NJ, USA # 566993), PE-conjugated Anti-CD62L Antibody (Beckman Coulter Life Sciences, Marseille Cedex, France #IM2214U), FITC-conjugated Anti-CD54 Antibody (Beckman Coulter Life Sciences, Marseille Cedex, France #IM0726U), KO525-A-conjugated Anti-CD31 Antibody (BD Horizon, Franklin Lakes, NJ, USA #563454), Alexa Fluor 488-conjugated Alkaline Phosphatase (BD Biosciences, Franklin Lakes, NJ, USA #561495), VWF Antibody (Santa Cruz Biotechnology, Santa Cruz, CA, USA #sc-53466), ACE Antibody (R&D Systems Minneapolis, MA, USA #AF929). For αSMA, VWF and ACE, cells were fixed and permeabilizated according to manufacturer protocol (Beckman Coulter Life Sciences Marseille Cedex, France #B31168). For the nonconjugated antibodies VWF and ACE, secondary FITZ-conjugated antibodies were used: Alexa Fluor 488-conjugated anti-rabbit (Jackson ImmunoResearch, Ely, UK #115-545-003) and Alexa Fluor 488-conjugated anti-goat (Jackson ImmunoResearch, Ely, UK #115-545-003), respectively.

### 4.4. Western Blotting

ECs cultured after 48 h in normoxia were washed with PBS and lysed in radio-immune precipitation assay (RIPA) buffer (Thermo Fisher Scientific, Rockford, IL, USA #89900) containing Protease Inhibitors Cocktail (Sigma-Aldrich, Darmstadt, Germany #P8340) and then incubated overnight at -80 °C. The same amounts of protein in the samples were assessed by BCA protein assay kit (Thermo Fisher Scientific, Rockford, IL, USA#23225) and then heated at 95 °C for 10 min. Next, separation was performed by sodium dodecyl sulfate (SDS)-polyacrylamide gel electrophoresis (SDS-PAGE) and transfer to nitrocellulose membranes (Bio-Rad, Hercules, CA, USA#1620094). After blocking nonspecific binding sites for 2 h using 5% nonfat dried milk in Tris-buffered saline/Tween at room temperature, membranes were incubated overnight at 4 °C with specific Abs: anti-p53 (1C12, Cell Signaling, Warsaw, Poland #2524), anti-ICAM-1 (Santa Cruz Biotechnology, Santa Cruz, CA, USA #sc-8439), anti-ZO-1 (Cell Signalling, Warsaw, Poland #5406), anti-VCAM-1 (Santa Cruz Biotechnology, Santa Cruz, CA, USA #sc-18864) and anti-Vinculin (V284, Santa Cruz Biotechnology, Santa Cruz, CA, USA #sc-59803). Then, incubation took place for 2h at RT with horse anti-mouse, goat anti-rabbit IgG or goat anti-rat secondary antibodies conjugated with horseradish peroxidase (HRP) (1:10,000; Vector Laboratories, Janki, Poland #PI-2000, #PI-1000, #PI-9400). Next, signals were detected by chemiluminescence substrate (Santa Cruz Biotechnology, Santa Cruz, CA, USA #sc-2048) on X-ray films (Carestream, Rochester, NY, USA #7711468). The density of bands was quantified by the ImageJ software (v.1.52p). Band intensities were normalized to the intensities of their corresponding loading controls (Vinculin, 117 kDa).

### 4.5. Next Generation Sequencing

Total RNA was isolated from the cell cultured in normoxia for 48 h, according to manufacturer’s protocol, with RNeasy Plus Mini Kit (Qiagen, Hilden, Germany#74136). Then, total concentration purity of the isolated material was evaluated using the fluorometer Qubit and Qubit RNA BR Assay Kit (Thermo Fisher Scientific, Singapore,#10210), according to manufacturers’ instructions. Samples were also investigated in order to assess the quality and integrity of RNA with Qubit RNA IQ Assay Kit (Thermo Fisher Scientific, Singapore, #33221). Next, NGS libraries were prepared with the NEBNext Library Prep Kit (BioLabs, Ipswich, MA, USA #E7770S). Finally libraries underwent quality assessment using the Bioanalyzer 2100 and High Sensitivity DNA Kit (Agilent Technologies, Santa Clara, CA, USA # 5067-4626), according to the manufacturers’ protocols. NGS assay were performed as an outsourced service. Differentially expressed genes (DEGs) were determined as those with *p* <  0.05 and fold change > 1.5. Functional enrichment analysis was performed using Cytoscape software (v.3.8.0) to identify gene ontology (GO) biological processes and Kyoto Encyclopedia of Genes and Genomes (KEGG) pathways represented by DEGs with statistical significance.

### 4.6. Secretion of VEGF-A

Both cell line’s media were changed to normoxic or hypoxic conditions (pO_2_ = 19% and pO_2_ = 1%, respectively) after overnight incubation in standard culture and previous incubation in the appropriate condition for 24 h. Next, cells were moved to culture in normoxia or hypoxia for 48 h. Then, media were collected and stored at −80 °C. ELISA was performed according to manufacturer protocol (R&D Systems, Minneapolis, MA, USA #DVE00) in three independent biological repetitions and three technical repetitions for each.

### 4.7. In Vitro Angiogenesis Assay

Angiogenesis was performed on Matrigel^TM^-coated 24-well plates (BD Biosciences, Franklin Lakes, NJ, USA). Cells were seeded at 5 × 10^4^ cells/cm^2^ and observed for 5 h in standard culture conditions using a Zeiss AxioObserver.7. The rearrangement of the cells and the formation of pseudo-vessels were followed for 5 h with a time step of 30 min at 5× magnification using Z1/7—software Zen v.2.6 (Zeiss, Oberkochen, Germany).

### 4.8. Permeability Test

Briefly, 0.4 nm filters (VWR, Warsaw, Poland #734-2746P) were coated with collagen IV (Sigma-Aldrich, Darmstadt, Germany #C752) and fibronectin (Sigma-Aldrich, Darmstadt, Germany #F1141) and left to solidify in sterile conditions. Then, HBH.MEC or HBCa.MEC, (1500 cells/well) were seeded on each filter placed into each well on a 96-well plate. Cells were cultured in 200 µL of normoxic medium for 48 h. Then fluorescein dye was added (Sigma-Aldrich, Darmstadt, Germany #46960-100G-F) and cells were incubated in standard culture conditions. After 30 min the measurement of fluorescence was measured at 488–520 nm.

### 4.9. Cell Proliferation Assay

In total, 1500 cells/well (HBH.MEC) and 2000 cells/well (HBCa.MEC) were seeded on 96-well plates in normoxia/hypoxia for 48 h. Then, CyQUANT Cell Proliferation Assay (Invitrogen, Rockford, IL, USA #C7026) was performed according to manufacturer protocol. Fluorescence of the samples was measured using a fluorescence microplate reader set up with 480 nm excitation.

### 4.10. Statistical Analysis

Statistical significance was calculated by Student’s *t*-test and two-way ANOVA, and p-value was calculated in GraphPad PRISM v.9.0. The data are represented by histograms and bar charts, each of which consisted of results from at least three independent experiments. For all experiments with error bars, the standard error of the mean (SEM) was calculated to indicate the variation within each experiment. *p* value of <0.05 was considered statistically significant and denoted with: **p* < 0.05, ***p* < 0.01, ****p* < 0.001 and *****p* < 0.0001.

## 5. Conclusions

Despite some limitations, our model offered a relevant tool for angiogenesis studies. Both cell lines, healthy ECs as well as pathological ECs, were obtained and cultured in normoxia— which is not a natural microenvironment. Some of the features, especially those of cancer ECs, might have been modified during prolonged culture, as indicated by exposure to hypoxia, which revealed further differences between cell lines. Nevertheless, breast TECs still possessed features of dysfunction in proangiogenic response and permeability. In this study, we demonstrated the properties of two cell lines deemed suitable for in vitro models of endothelial organospecific cells reflecting the phenotypes of healthy and tumor tissues. This entire characterization indicated that HBH.MEC and HBCa.MEC provide a valuable in vitro model of breast tumor angiogenesis, permeability and leukocyte rolling studies which mimic cell behavior and (dys)function in pathological vessels of the most lethal breast cancer subtype. Furthermore, it confirmed the validity of the endothelial organospecificity for the design of biologically relevant tumor models.

## Figures and Tables

**Figure 1 ijms-22-08862-f001:**
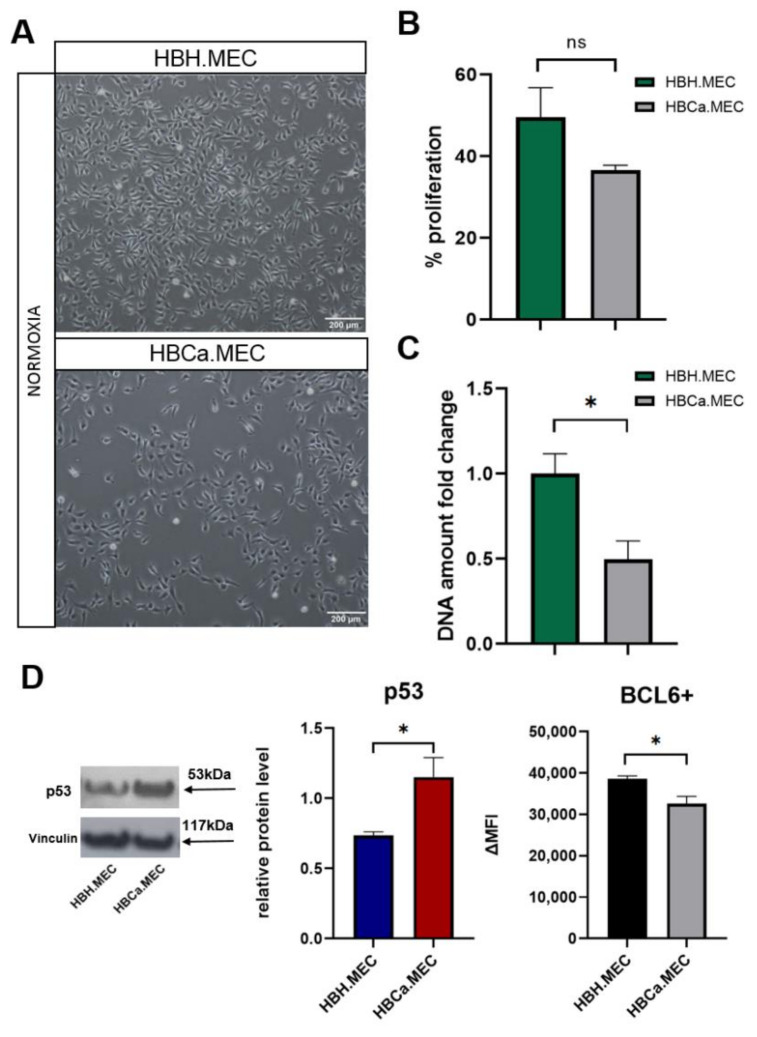
The influence of new endothelial cell lines phenotype on morphology, viability and apoptosis induced proteins. (**A**) Cell density after 48 h in normoxia. Cells were seeded at t = 0 h in the same number. Magnification 4×. (**B**) The influence of ECs pathological state on cell proliferation. Cells were seeded at 0 h in the same number and cultured in normoxia for 48 h. (**C**) DNA proliferation confirmed that tumor-derived ECs multiplied slower in normoxia when compared to healthy ECs. * *p* < 0.05 in Student *t* test vs. HBH.MEC normoxia (**D**) The protein level of BCL6 and p53. Cells were stained for BCL6 and undergo flow cytometry analysis. Data were recorded for 10,000 events using CellQuest software (v.2.3.0.84) and presented as delta MFI. The level of p53 (53 kDa) was evaluated on WB, relatively to loading control, Vinculin (117 kDa). Representative bands are shown. Bar chart presents data from ImageJ analysis (v.1.52p). Data are reported as the means ± SEM (n = 3). Ns—not significant; * *p* < 0.05 in Student *t* test vs. HBH.MEC.

**Figure 2 ijms-22-08862-f002:**
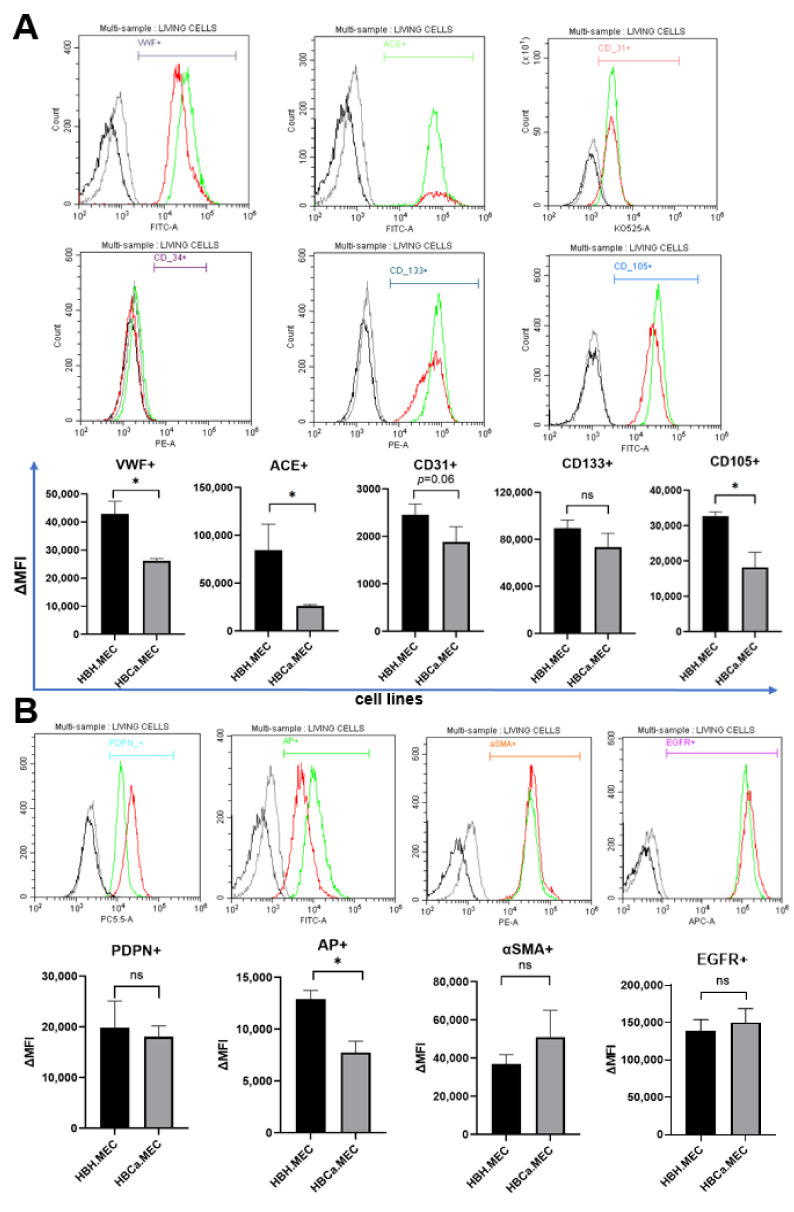
Characterization of ECs by flow cytometry markers. The cells were stained for the following markers: VWF, ACE, CD31, CD34, CD133, CD105, PDPN, AP, αSMA, EGFR. Data were recorded for 10,000 events using CellQuest software (v.2.3.0.84) and presented as histogram overlays. (**A**) The most characteristic markers for ECs and EPCs. (**B**) Markers associated with cancerous phenotype. Histogram overlays display representative repetitions; gray—healthy ECs unstained, black—tumor ECs unstained, green—healthy ECs stained, red—tumor ECs stained. Y axis = the number of events; X axis = fluorescence intensity; the bar charts present delta MFI. Ns—not significant; * *p* < 0.05 in Student *t* test vs. HBH.MEC. Data are reported as the means ± SEM (n = 3).

**Figure 3 ijms-22-08862-f003:**
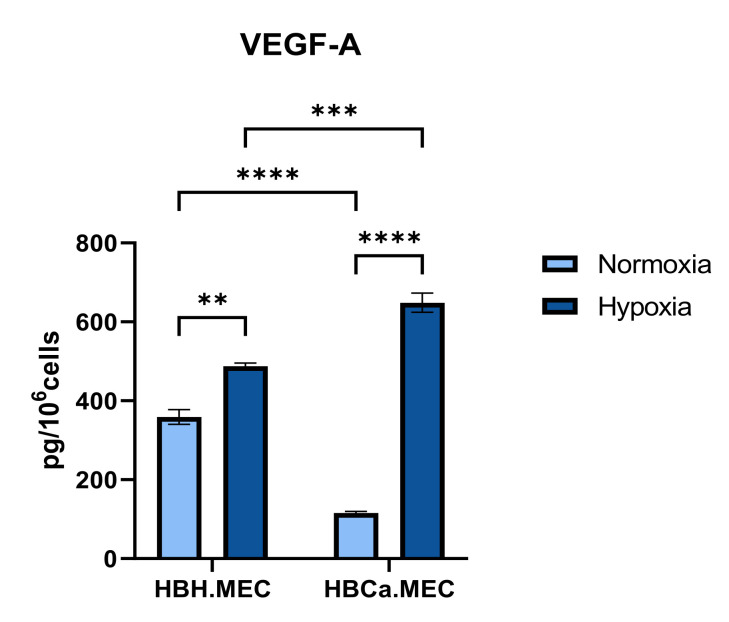
Secretion of VEGF-A produced by HBH.MEC and HBCa.MEC cultured upon normoxic and hypoxic conditions. Supernatants from both cell lines were collected after culture for 48 h in normoxia (19% O_2_) or in hypoxia (1% O_2_). Secretion of VEGF-A was evaluated by using the ELISA. Results are expressed as pg/million cells ± SEM, ** *p* < 0.01, *** *p* < 0.001, **** *p* < 0.0001 in two-way ANOVA (n = 3).

**Figure 4 ijms-22-08862-f004:**
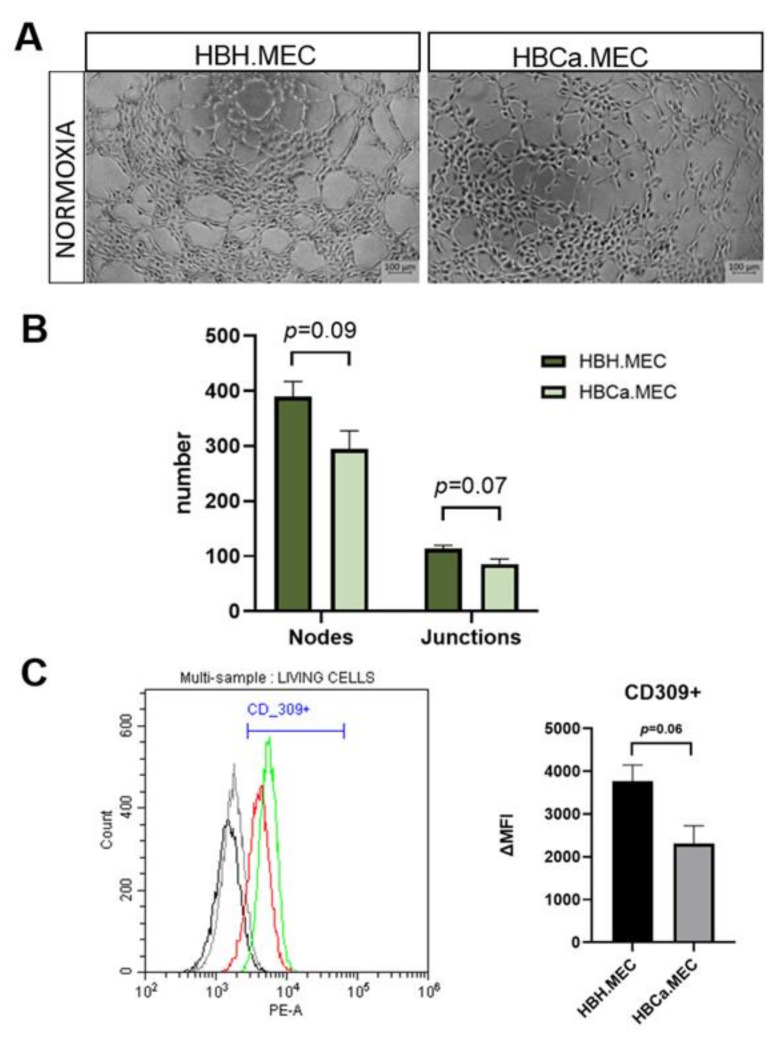
The effect of pathological state of ECs on the angiogenic potential. (**A**) Morphology of tubes formed in vitro by HBH.MEC/HBCa.MEC after 5 h in normoxia, magnification 5× (representative pictures). (**B**) The number of nodes, junctions was estimated by Image J software (v.1.52p). *p* < 0.05 in Student *t* test vs. HBH.MEC. Data are reported as the means ± SEM (n = 3). (**C**) The level of CD309 in both cell lines. The cells were stained for CD309 and data were recorded for 10,000 events using CellQuest software (ver.2.3.0.84) and presented as histogram overlay (shows representative repetitions; gray—healthy ECs unstained, black—tumor ECs unstained, green—healthy ECs stained, red—tumor ECs stained. Y axis = the number of events; X axis = fluorescence intensity). The bar chart presents delta MFI. *p* = 0.06 in Student *t* test vs. HBH.MEC. Data are reported as the means ± SEM (n = 3).

**Figure 5 ijms-22-08862-f005:**
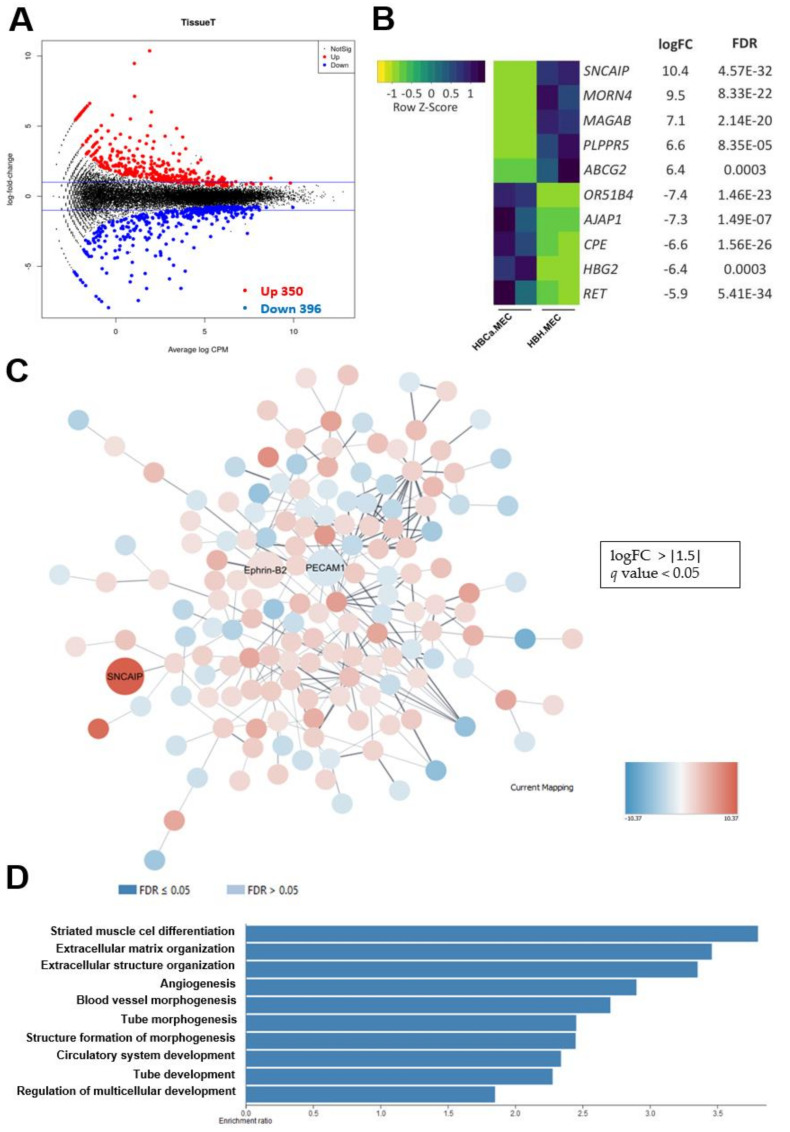
Differences in the transcriptome profiles of tumor-derived ECs and healthy ECs. (**A**) Volcano plot of significantly differentially expressed genes of pathological ECs vs. healthy ECs. (**B**) top 10 of the most differentially expressed genes of pathological ECs vs. healthy ECs; FDR-false discovery rate. (**C**) Functional enrichment network performed in Cytoscape software (v.3.8.0). (**D**) Enrichment analysis based on Gene Ontology biological processes shows the top 10 activated processes in both cell types (web-based gene set analysis toolkit enrichment method: ORA; organism: homo sapiens, enrichment categories: geneontology biological_process).

## Data Availability

All data generated or analyzed during this study are included either in this article or in the Appendix A. The data that support the findings of this study are available from the corresponding author kwilkus@wim.mil.pl upon reasonable request. NGS data are available at GEO database (7 July 2021) https://www.ncbi.nlm.nih.gov/geo/query/acc.cgi?acc=GSE179509.

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
