# Peer review of "Distinctive Properties of Endothelial Cells from Tumor and Normal Tissue in Human Breast Cancer"

_ijms, 2021, doi:10.3390/ijms22168862_

Round 1
Reviewer 1 Report
This is a really interesting study by Wilkus and colleagues who have submitted a body of work on the phenotypic and function differences between endothelial cells harvested from human breast cancer tissue versus healthy tissue from the same donor.
In my opinion, the authors have presented some potentially novel and exciting observations that provide a unique insight into the effects of cancer cells on the endothelial cells of the blood vasculature. While this study addresses some diveres and interesting points, I have some concerns related to the data presented and provide some suggestions to improve the study.
Recommendations are listed below:
- The manuscript would benefit from some editorial assistance to improve the English grammar and intended context of the wording. The abstract in particular lacks clarity in the message it aims to deliver. Outsourcing to a professional organisation for assistance would address this.
- Even though this has been previously published, for ease of the reader, please provide more detailed information on the patient from whom these cells were harvested and immoratlised.
- Include that KDR is VEGFR2
- Fig 1, undertake more experiments to definitely determine whether the HBCa.MEC cells have a slower proliferative rate and express more p53. Show a immunoblot on which the bands of interest are run next to eache other, not spliced from across the same gel/transfer. This also applies to the Figure S1.
- All flow cytometry histograms should be presented with a y-axis of ‘% max’ not counts. This also applies to the Figure S1.
- Fig 4A needs to be quantified for tube numbers, tube length and loops (as is the current convention), from multiple experiments.
- Fig 5, please indicate the gene expression levels for the proteins identified in the figures above to reveal whether the changes observed in protein levels correlate with gene differences as well. Statistical differences also need to be included here.
- Finally, to really consolidate/validate this work, the differences observed between the ‘healthy’ and ‘cancerous’ ECs should be investigated further by culturing the ‘healthy’ ECs with supernatant from BrCa cells to determine whether any of the aforementioned differences (e.g. CD54, CD309, VEGFA, p53, vWF, AEC, CD105) are altered in response to a soluble (secreted) factor generated by the cancer cells themselves.
Author Response
Dear Reviewer,
We thank for the excellent critique and feedback. We tried to respond to the suggestions and we hope that these revisions improved the manuscript significantly. Below, please find the point-by-point responses to the reviews and we submitted the manuscript in the track changes mode to allow to follow the modifications made in the text.
Please see the attachment.
Yours faithfully,
Kinga Wilkus
Laboratory Of Molecular Oncology and Innovative Therapies Military Institute of Medicine Szaserów 128 04-141 Warsaw, Poland

Reviewer 2 Report
The present article titled “Distinctive properties of endothelial cells from tumor and normal tissue in human breast cancer” is focused on the in vitro characterization and comparative analysis (on molecular and functional levels) of normal ECs isolated from healthy breast tissue as well as pathological ECs (tumor-derived endothelial cells). In my opinion it’s a quite interesting project that can help better understand and mimic the cancer microenvironment. Well done!
Author Response
Dear Reviewer,
We would like to thank for appreciation. We hope that presented model will be of value for other researchers and will contribute to the development of knowledge in the field of angiogenesis. We will continue using these cell lines in more detailed molecular experiments and develop other organospecific endothelial lines.
Yours faithfully,
Kinga Wilkus
Laboratory Of Molecular Oncology and Innovative Therapies Military Institute of Medicine
Szaserów 128
04-141 Warsaw, Poland